ancestry; cardiovascular disease; genetics; therapeutics; precision medicine

**Author for correspondence:**
Dipender Gill,
E-mail: dipender.gill@imperial.ac.uk

O.S. and M.-J.D. contributed equally and are joint first authors.

# Genetic heterogeneity in cardiovascular disease across ancestries: Insights for mechanisms and therapeutic intervention

Opeyemi Soremekun[1,2,3], Marie-Joe Dib[2,4], Skanda Rajasundaram[5,6], Segun Fatumo[1,7] and Dipender Gill[2,4] 

[1]The African Computational Genomics (TACG) Research Group, Medical Research Council/Uganda Virus Research Institute and London School of Hygiene and Tropical Medicine, Uganda Research Unit, Entebbe, Uganda; [2]Department of Epidemiology and Biostatistics, School of Public Health, Imperial College London, London, UK; [3]Molecular Bio-Computation and Drug Design Laboratory, School of Health Sciences, University of KwaZulu-Natal, Westville Campus, Durban, South Africa; [4]British Heart Foundation Centre of Excellence, Imperial College London, London, UK; [5]Centre for Evidence-Based Medicine, University of Oxford, Oxford, UK; [6]Faculty of Medicine, Imperial College London, London, UK and [7]Department of Non-Communicable Disease Epidemiology (NCDE), London School of Hygiene and Tropical Medicine, London, UK

## Abstract

Cardiovascular diseases (CVDs) are complex in their aetiology, arising due to a combination of genetics, lifestyle and environmental factors. By nature of this complexity, different CVDs vary in their molecular mechanisms, clinical presentation and progression. Although extensive efforts are being made to develop novel therapeutics for CVDs, genetic heterogeneity is often overlooked in the development process. By considering molecular mechanisms at an individual and ancestral level, a richer understanding of the influence of environmental and lifestyle factors can be gained and more refined therapeutic interventions can be developed. It is therefore expedient to understand the molecular and clinical heterogeneity in CVDs that exists across different populations. In this review, we highlight how the mechanisms underlying CVDs vary across diverse population ancestry groups due to genetic heterogeneity. We then discuss how such genetic heterogeneity is being leveraged to inform therapeutic interventions and personalised medicine, highlighting examples across the CVD spectrum. Finally, we present an overview of how polygenic risk scores and Mendelian randomisation can foster more robust insight into disease mechanisms and therapeutic intervention in diverse populations. Fulfilment of the vision of precision medicine requires more exhaustive leveraging of the genetic variability across diverse ancestry populations to improve our understanding of disease onset, progression and response to therapeutic intervention.

## Impact statement

This review discusses how the genetic basis of cardiovascular disease (CVD) can differ across different ancestries. It focuses on common CVDs such as coronary artery disease (CAD), stroke and their modifiable risk factors (body mass index, type 2 diabetes mellitus, high cholesterol and high blood pressure). It describes how genetic differences, or heterogeneity, can lead to different molecular mechanisms driving CVD across different ancestries. It then discusses how such heterogeneity could be used to improve the early diagnosis of CVD and inform the development of new CVD therapies. For instance, disease mechanisms potentially independent of atherosclerosis may drive CAD in East Asian populations, whereas certain molecular mediators may represent therapeutic targets for stroke that are specific to African ancestry individuals. The review provides insight for researchers, clinicians, funders and healthcare policymakers to understand the importance of genetic heterogeneity across ancestries in the prevention, prediction and treatment of CVD. It highlights instances of genetic ancestry influencing an individual's response to cardiovascular medication and argues that the practice of precision medicine requires a greater understanding of such influences. Although focused on CVD, the content will pertain to many other disease areas and will be of interest to anyone involved in the application of genomics to clinical medicine.

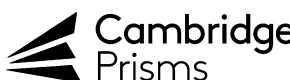



## Introduction

Over the last decade, the field of genetics has rapidly advanced and contributed to understanding of pathogenic mechanisms underlying rare and complex diseases (Gurdasani et al.,

2019). Genetics is also increasingly leveraged to successfully identify novel drug targets (Ochoa et al., 2022). Genome-wide association studies (GWASs) and next-generation sequencing methods have been successful in identifying risk loci for disease. However, most of these studies have predominantly been conducted in populations of European ancestry (Fatumo et al., 2022). In fact, as of 2021, 86% of genomic studies have been carried out in individuals of European descent (Fatumo et al., 2022). Disparities in health status and outcomes between different ancestry populations are increasing, and a lack of diversity in genetic research may exacerbate inequities (Bentley et al., 2017). Leveraging genetic data from ancestrally diverse populations can provide deep insights into specific pathogenic variants that differ across population groups (Bentley et al., 2017; Martin et al., 2017; Atutornu et al., 2022).

The genetic basis of disease is classified into Mendelian disorders and complex polygenic diseases. Mendelian diseases are caused by single gene alterations, whereas complex diseases are caused by several genes, each of which play a small, additive role in increasing disease risk. Irrespective of the relative contribution of genetics and environment to traits and diseases, 'heterogeneity' is observed in disease outcomes and plays a role in their mechanism (Bomprezzi et al., 2003). Woodward et al. (2022) define heterogeneity as disparities that exist at different taxonomic levels such as cells, tissues and phenotypes. Some of these disparities could either be directly accounted for, and are measurable, whereas others are not (Woodward et al., 2022). In population health, heterogeneity is an inevitable phenomenon that pertains to numerous epidemiological concepts, including disease aetiology, missing heritability and treatment resistance (Woodward et al., 2022). It is important to address this to facilitate tailored disease interventions in multi-ancestry and admixed populations.

In this review, we highlight findings from genetic research of cardiovascular disease (CVD) outcomes: coronary artery disease (CAD) and stroke, and their modifiable risk factors (body mass index [BMI], type 2 diabetes mellitus [T2DM], lipids and hypertension in diverse ancestry population groups). These risk factors were prioritised on the basis of their significant contribution to CVD burden across diverse populations (Yusuf et al., 2020; Fawzy and Lip, 2021; Shah et al., 2021; Wang et al., 2021). Rather than systematically search through all previously published work in this area, we instead prioritise studies that highlight the insights that can be gained by an appreciation of genetic heterogeneity across populations of diverse ancestry groups. Such understanding has the potential to improve our overall ability to assess disease risk, and thus serves as a conduit between precision medicine and public health for improving the well-being of individuals and populations (Wehby et al., 2018; Roberts et al., 2021) (Figure 1).

### Genetic heterogeneity

Genetic heterogeneity can be defined as genetic variation that results in the same (or similar) phenotype(s) (Woodward et al., 2022), where a phenotype is an organism's set of observable characteristics or traits. Genetic heterogeneity can contribute substantially to complex disease phenotypes. Two different types of genetic heterogeneity are well described in the literature: allelic heterogeneity, which arises when different alleles at the same genetic locus result in the same phenotype, and locus heterogeneity, which arises

when mutations in different loci lead to the same phenotype (Scriver, 2006; Woodward et al., 2022).

### Genetic heterogeneity and ancestral differences in CVD outcome and risk factors

Much attention is currently being paid to how genetic factors may contribute to disparities in health and disease, although the limitations of commonly used ethnic descriptors in explaining the genetic structures in diverse populations have been discussed (Wilson et al., 2001). This begs the question of how race is defined within the scope of genetics. In population studies, the terms 'race' and 'ethnicity', which take into account cultural, linguistic, biological and geographical aspects, are frequently used interchangeably (Sankar and Cho, 2002), and they also allude to an individual's phenotypic features (Peterson et al., 2019). Ancestry connotes an individual genetic ancestry as evidenced by the DNA passed down and through generations in a specific group (Peterson et al., 2019).

We highlight below current progress in the genetic study of CVD aetiology for diverse ancestry groups. We prioritised CAD, stroke and modifiable risk factors (BMI, T2DM, lipids and hypertension) in selected ancestry groups. While other CVD endpoints and modifiable risk factors, these were selected according to their burden and contributions to CVD (Wehby et al., 2018; Roth et al., 2020).

### *Coronary artery disease*

CAD is a common polygenic disease, and is a leading cause of morbidity and mortality globally (GBD 2015 Mortality and Causes of Death Collaborators, 2016). CAD typically causes myocardial ischaemia due to narrowing or blockage of the coronary arteries that feed into heart, leading to myocardial infarction. Development of arrhythmia, heart failure and death are also observed consequences of CAD. CAD is known to be highly heritable (Marenberg et al., 1994) with more than 160 CAD susceptibility loci described (Nikpay et al., 2015; van der Harst and Verweij, 2018). Advances in the field of genetics have not only revealed novel CAD disease pathways, but have also enabled the quantification of individual genetic risk and the development of new therapeutic agents (Miyazawa and Ito, 2021). In a recent GWAS for CAD (Koyama et al., 2020), Koyama et al. used the WGS data of 4,930 Japanese individuals and created a reference panel containing disease-specific haplotype (physical grouping of genomic variants usually inherited together) information for 1,782 patients with CAD for imputation. They identified an association between CAD and a missense mutation in *RNF213*, which has been reported as a causative gene of Moyamoya disease. Here, the genetic investigation of disease via WGS and GWAS efforts in a Japanese population revealed a common genetic risk factor between CAD and Moyamoya disease, providing novel mechanistic insight into CAD. More specifically, this offered insight into the pathological features of Moyamoya disease in relation to atherosclerosis (Houkin et al., 2012). Findings from the Japanese populations have provided evidence of disease mechanisms for CAD potentially separate to that of atherosclerosis, thereby highlighting the heterogeneity in disease mechanisms underlying CAD. It remains to be studied whether the mechanism is specific to East Asian populations or if this translates to other ancestries.

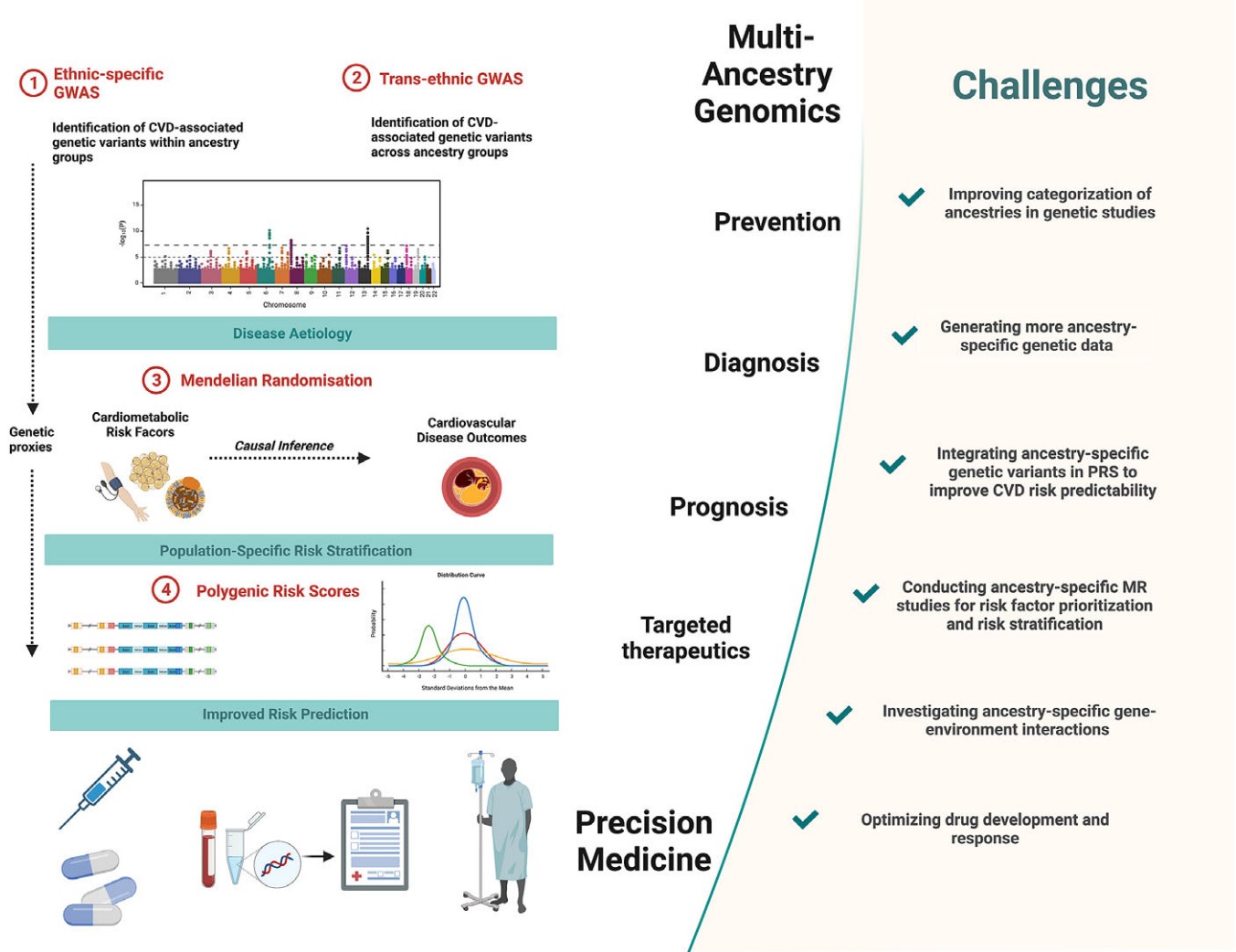

**Figure 1.** Precision medicine approaches in cardiovascular disease (CVD) and challenges to overcome. Multi-ancestry genetic studies play a pivotal role in advancing precision medicine. Comparisons of ancestry-specific and trans-ancestry GWAS findings provide insights into CVD aetiology and its heterogeneity. Secondary analyses of GWAS data, notably using Mendelian randomisation methods, provide additional insights into causal relationships between cardiometabolic risk factors and CVD outcomes. This allows for risk factor prioritisation and optimised risk stratification in diverse ancestry population groups. Integrating ancestry-specific GWAS associations in polygenic risk scores allows for improved predictability of CVD outcomes. Together these approaches contribute to the improved primary prevention, diagnosis and prognosis and targeted therapeutics of CVD. Figure created using BioRender.com (2019).

## Stroke

Stroke is another leading cause of disability and death worldwide, exerting a significant strain on healthcare systems (GBD 2017 Causes of Death Collaborators, 2018). There are considerable disparities in stroke incidence, subtype and prognosis between those of European and African ancestries, with established stroke risk facts explaining only about half of the variation (Prapiadou et al., 2021). In some studies conducted in the United States, African ancestry individuals aged between 45 and 64 years have a threefold higher risk of stroke compared with non-Africans (although this difference is attenuated by age 85) (Rosamond et al., 1999; G. Howard et al., 2011; V. J. Howard et al., 2011). The disparity observed in this study is ascribed to an increased incidence in African ancestry individuals rather than decreased survival. While many studies have documented interracial differences in the incidence of stroke, the reasons for these differences have not been fully explained, and

therefore the identification of ancestry-specific risk factors is important for the treatment and management of stroke. In a study by Harriott et al. (2015), the rs11172113 variant, which mapped on to the *LRP1* gene, was associated with stroke among African Americans, but this result failed to replicate in a non-Hispanic White cohort. *LRP1* plays a key role in the liver by removing atherogenic lipoproteins and other proatherogenic ligands from circulation (Chen et al., 2021). Anti-P3 (Gly1127-Cys1140) antibodies (Abs) that block the LRP1 (CR9) domain have been demonstrated to stop LRP1-mediated aggregated-LDL (aggLDL) internalisation and aggLDL-induced *LRP1* upregulation, preventing foam cell formation in human macrophages and vascular smooth muscle cells (Costales et al., 2015; Bornachea et al., 2020). The strong link between *LRP1* and stroke via the atherosclerotic pathway renders *LRP1* as a potential therapeutic target for stroke in this population. The multi-ethnic Stroke Prevention in Young

Women case–control research discovered two SNPs in the *NOS3* gene that were related to ischemic stroke in African ancestry women but not in European women (T. D. Howard et al., 2005). *NOS* has also been reported to be associated with stroke in a Chinese population (Hou et al., 2001; Berger et al., 2007) without been replicated in a Japanese population (Yahashi et al., 1998). *NOS3* catalyses the production of nitric oxide, which is responsible for mediating vascular relaxation in response to vasoactive substances and stress (Förstermann and Sessa, 2012). *NOS3* inhibits platelet aggregation and suppresses smooth muscle proliferation. Therefore, NOS3's properties make it a biologically plausible candidate to investigate as a susceptibility gene in ischemic stroke for particular population groups.

### BMI

Obesity has been linked to an increased risk of noncommunicable diseases, including T2DM (Boles et al., 2017) and CVD (Ortega et al., 2016), and is used as a proxy for obesity. Genetics and environmental factors are known to influence BMI in individuals (Bhaskaran et al., 2014). Elevated BMI predisposes individuals to numerous diseases (Bhaskaran et al., 2014; Benjamin et al., 2017), and BMI heritability is estimated to be approximately 40% (Hemani et al., 2013; Yang et al., 2015). GWAS of BMI has identified up to 426 BMI loci (Liu et al., 2008; Thorleifsson et al., 2009; Willer et al., 2009; Speliotes et al., 2010; Kim et al., 2011; Turcot et al., 2018). Notable allele frequency distribution has been observed in BMI-associated variants which consequently confers ancestral differences in BMI. For example, in a trans-ancestry meta-analysis by Downie et al. (2022), wide variation of minor allele frequency was seen across populations for sentinel variants ranging from 0.12 to 0.36 for *VEGFA* locus and 0.10 to 0.37 for *PTEN* locus. A population specific locus (*LRRC37A5P*) was found in individuals that self-identify as African Americans (Downie et al., 2022). The identification of these ancestry specific loci underscores the significance of undertaking genetic studies in diverse populations. The metabolic effects of obesity have been linked to the biological activity of adipose tissue in a manner specific to fat distribution, such as visceral and subcutaneous adiposity or fat accumulation, which can vary substantially across populations (Crowther et al., 2006).

### Type 2 diabetes mellitus

More than 200 genetic variants have been found through GWAS to be associated with T2DM across various populations (Mahajan et al., 2018). Most have small effects on diabetes risk, but a few have larger effects across different ancestral populations. A study has found that allele frequencies for established T2DM susceptibility variants differ significantly across ancestry groups, with African ancestry groups having the highest genetic risk, East Asians and American Indians having the lowest genetic risk and Europeans having an intermediate risk (Klimentidis et al., 2011). Using genome-wide SNP data from the Human Genome Diversity Panel of 938 individuals from 53 populations, Klimentidis et al. (2011) compared the population differentiation and haplotype pattern of genome-wide significant genes and the rest of the genome. East Asians and sub-Saharan Africans differ the most in terms of differentiation, implying that T2DM-associated genes in these populations have been subject to increased selection pressures. When compared with sub-Saharan Africans and Native Americans, haplotype analysis indicates an excess of obesity loci with signs of recent positive selection among South Asians and Europeans

(Klimentidis et al., 2011). The authors of the study suggested that genetic regions around loci driving T2DM have undergone substantial evolutionary changes and selection in the last 100,000 years, most notably in sub-Saharan Africans and East Asians. Therefore, the identification of loci that have undergone this recent selection may be useful in teasing out population-specific risk variants for T2DM treatment. Using a meta-regression model which allows for the description of heterogeneity based on ancestry, environmental factors or study design, Mahajan et al. (2022) explored the effect of heterogeneity in diverse ancestral populations. They found 136 loci associated with T2DM to be driven by ancestral heterogeneity and 27 loci driven by study design or environmental exposures. From these findings, it is suggested that the heterogeneity in effect sizes observed across different ancestral populations is due to genetic variation more than study design or geographical location (Mahajan et al., 2022). In another study by Chen et al. (2012), disease association data from 5,065 papers were manually curated, and T2DM genetic risk was seen to be higher for individuals in the African populations and lower in the Asian populations. Some ancestry-specific gene–environment interaction factors may be responsible for the disparity observed; hence, further GWASs adjusting for many environmental factors could help understand the mechanisms and origin of T2DM across different ancestries. T2DM has been linked to changes in beta-cell activity and reduced insulin production (Haffner et al., 1996; Ferrannini and Mari, 2004; Lorenzo et al., 2010). Although the molecular mechanisms underlying altered beta-cell secretion and insulin kinetics in T2DM patients are unknown, there is clear evidence for genetic (and epigenetic) as well as environmental factors such as physical inactivity and overweight/obesity, which are more prevalent in Africans and Europeans (Kolb and Martin, 2017; Ali et al., 2018; Dendup et al., 2018) (Table 1).

### Lipids

High levels of circulating low-density lipoprotein cholesterol (LDL-c) and low levels of circulating high-density lipoprotein cholesterol (HDL-c) are risk factors for stroke and heart disease (Roger et al., 2011). Of note is the opposing relationship of LDL-c with ischemic and haemorrhagic stroke in Chinese populations (Sun et al., 2019), which highlights the need for careful phenotypic definitions when ascertaining the role of genetic variation across studies considering different populations.

Blood lipid levels, including LDL-c, HDL-c and triglycerides (TG), are heritable, with known genetic variants explaining 10%–15% of phenotypic variations (Pilia et al., 2006). Evaluation of transferability of lipid associations detected in a European discovery GWAS to Asian and African ancestry replication cohorts shows considerable variation in the extent of replication of the three lipid traits (Kuchenbaecker et al., 2019). While more than 75% of variants with strong associations ($P$-value $< 10^{-100}$) for HDL-c and LDL-c replicate in all ancestries, only approximately 57% of strong TG associations replicate in the African cohort. Moreover, the associations detected at higher $P$-values showed much lower transferability (<30% in African populations across lipid traits). Although the transferability of associations to African populations might improve substantially with the use of more trans-ethnic discovery GWASs and larger representative African datasets, there is a strong possibility a sizable portion of these associations might be actually ancestry-specific (Choudhury et al., 2022). For example, the largest multi-ancestry GWAS by Graham et al. (2021) showed that 76% of the 773 lipid associated regions identified in at least one

**Table 1.** Examples of genes implicated in different CVD outcomes and risk factors that confer heterogeneity across ancestries

| Traits | Genes | References |
|---|---|---|
| Type 2 diabetes | *HNF1A, TBC1D4, IRS1, ADAMTS9, ARL15, ZFAND3, PTPRD, TCF7L2, MPHOSPH9, C2CD4A, SLC16A11, DUSP9, AGTR1, IL6, NOS3, TNFA* | Hanson et al. (2015); Golden et al. (2019); Shoily et al. (2021) |
| Lipids | *GCKR, NRXN3, TTC7B, LPL, LIPC, CETP, PON1, APOE, NOS3* | Chang et al. (2010); Ellman et al. (2015); Shetty et al. (2015) |
| Stroke | *COL4A1, PDE3A, CDKN2B* | Chauhan and Debette (2016); Kamin Mukaz et al. (2020); Surakka et al. (2022) |
| Hypertension | *C10orf107, SH2B3, DPEP1, CACNB2, ALDH2* | Takeuchi et al. (2018) |
| BMI | *VEGFA, PTEN, LRRC37A5P* | Downie et al. (2022) |
| CAD | *APOB, FN1, ATF6B, HDAC9, UBE3B, RPH3A, ADAMTS7, ABO* | Matsunaga et al. (2020) |

Abbreviations: BMI, body mass index; CVD, cardiovascular disease.

of the five ancestries studied were found in Europeans, 15 loci were unique to Admixed African or Africans, 6 to East Asian, 6 to Hispanics and 1 to South Asians.

## Hypertension

Hypertension is a major risk factor for CVD with an estimated heritability between 30% and 60% (Sung et al., 2018), and more than 200 genetic loci are known to be related with hypertension (Ehret et al., 2011, 2016; Surendran et al., 2016). The risk of developing hypertension is attributable to genetic, environmental and demographic factors. The prevalence of hypertension is higher in individuals of East Asian ancestry, who also have a higher risk of stroke than their European counterparts (Takeuchi et al., 2018). To determine if heterogeneity exists in BP traits between East Asians and Europeans, Takeuchi et al. performed a multi-staged GWAS. In this study, they found inter-ancestry heterogeneity in eight loci mapped near *CACNB2, C10orf107, SH2B3, DPEP1* and *ALDH2* (Takeuchi et al., 2018). *ALDH2* is an important enzyme involved in alcohol metabolism. The polymorphism induced by rs671 produces an inactive subunit of *ALDH2*, which leads to accumulation of acetaldehyde after alcohol intake (Takeshita et al., 1993). Acetaldehyde elevation lowers blood pressure through vasodilation, which is linked to the characteristic physiological effects such as high temperature, increased heart and respiration rates and palpitations seen among ALDH2 *2*2 homozygotes, the frequency of which varies across different ancestry groups (Quertemont and Didone, 2006).

## Clinical importance of leveraging genetic heterogeneity across ancestry groups

An appreciation of genetic heterogeneity across ancestries has a number of advantages. First, it would improve our understanding of disease mechanisms given that pathophysiology likely varies across ancestries in part due to genetic variation. For instance, selectively studying those genes that are ancestry-specific would shed more light on the pathogenesis and clinical presentation of CVD. Second, elucidating ancestry-specific molecular pathways involved in a disease can in turn help determine ancestry-specific susceptibility to the disease. Different ancestries carry different combinations of risk alleles that predispose them to disease risk. Identifying how these risk alleles vary across ancestries would help in early detection of individuals at high risk and further help prioritise those individuals who would benefit from intervention.

Third, understanding how genetic heterogeneity influences an individual's response to drugs is important, given that many drugs are primarily developed in European ancestry individuals. Finally, knowledge of differential susceptibility to risk factors improves clinical management of patients. For instance, in the prevention of stroke, blood pressure control may be more important in African ancestry individuals given that the risk of stroke in African ancestry individuals with hypertension is three times higher than that of Europeans (Spence and Rayner, 2018).

## Differences in drug response

Ancestry can influence inter-individual differences in drug exposure and/or responsiveness, altering the risk–benefit ratio in certain subgroups of patients (Figure 2). Differences in drug responsiveness between different ancestries may in part be attributable to differences in the distribution of polymorphisms associated with the enzymes involved in drug metabolism. A single-nucleotide variation in a candidate gene can have a significant impact on pharmacological response (Cazzola et al., 2018). Individuals of different ancestries have been shown to respond differently to antihypertensive therapy (Preston et al., 1998; Julius et al., 2004; Wright et al., 2005; Shin and Johnson, 2007; Gong et al., 2016), heart failure therapy (Carson et al., 1999; Beta-Blocker Evaluation of Survival Trial Investigators et al., 2001; Exner et al., 2001; Dries et al., 2002), lipid-lowering therapy (Lee et al., 2005; 'High-dose atorvastatin after stroke or transient ischemic attack', 2006; Liao, 2007; Link et al., 2008; Ieiri et al., 2009; SEARCH Collaborative Group, 2010; Hu et al., 2012; H.-K. Lee et al., 2013), antiplatelet therapy (Mega et al., 2010; Chan et al., 2012) and anticoagulant therapy (You et al., 2005; Keeling et al., 2011; Hori et al., 2013; Yamashita et al., 2015) (Table 2).

## Limited transferability of polygenic risk scores across diverse population groups

Risk prediction of cardiometabolic traits and CVD through genetic risk scores may be more clinically applicable through an enhanced understanding of the genetic architecture of complex traits, population risk-stratification and tailored interventions (Márquez-Luna et al., 2017). The use of European data for polygenic risk score (PRS) prediction in non-European and genetically diverse populations reduces prediction accuracy due to ancestral differences in LD patterns and allele frequencies. The lack of PRS optimised for non-European populations is a substantial obstacle in paving the

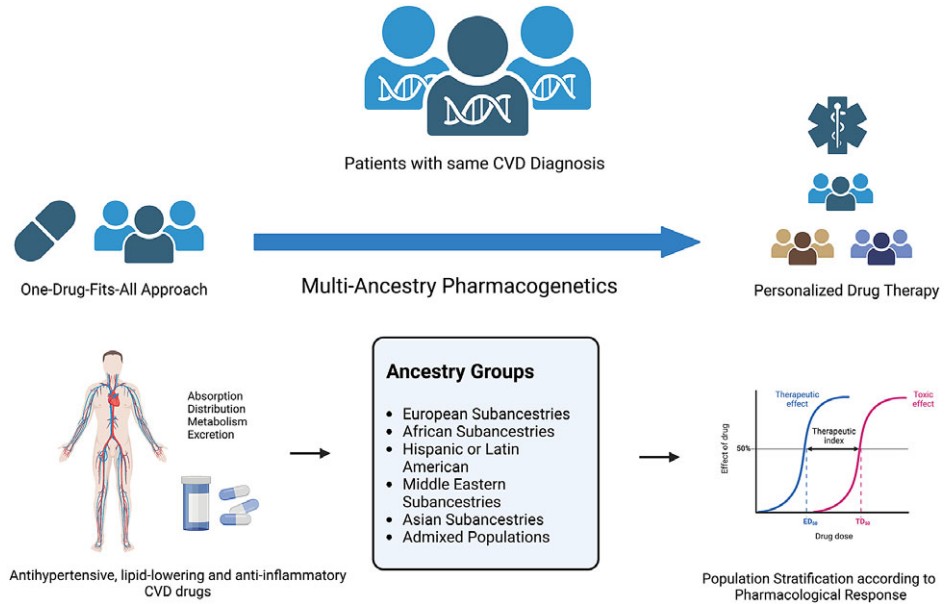

**Figure 2.** Multi-ancestry pharmacogenetics in the scope of personalised drug therapy. Figure created using BioRender.com.

**Table 2.** Influence of ancestry on drug response

| Drug class | Drug | Ancestral differences | References |
|---|---|---|---|
| Antihypertensive | Lisinopril | Increased risk of stroke in African ancestry individuals, whereas no such effect was seen in non-Africans. | Wright et al. (2005) |
| | Beta-blockers and angiotensin-converting-enzyme (ACE) inhibitors | African ancestry individuals have poorer BP lowering response compared with non-Africans. | Seedat and Parag (1987); Cohn et al. (2004) |
| | Calcium channel blockers (CCBs) | African ancestry individuals showed better response to CCBs when compared with non-Africans. | Nguyen et al. (2009) |
| | ACE | African ancestry was associated with reduction in SBP and DBP response to ACE | Peck et al. (2013) |
| | Isosorbide-hydralazine (I-H) | African ancestry individuals benefited more from I-H than non-Africans. | Carson et al. (1999) |
| Lipid lowering | Statins | Higher level o-expression of *OATP1B1* and *OATP1B3* in Asians compared with Europeans. | Peng et al. (2015) |
| | Pravastatin | Higher AUC and Cmax was seen in European-Americans compared with African-Americans. | Ho et al. (2007) |
| | Propranolol | 53%–76% clearance in Africans compared with Europeans. | Sowinski et al. (1996) |

way in the roadmap to precision diagnostics (Fatumo et al., 2022). Using multi-ancestry summary statistics has the potential to enhance PRS performance in diverse populations, as demonstrated for CAD. Conducting trans-ancestry meta-analyses helped discover 35 additional new CAD loci, which enabled the construction of a PRS for CAD that outperformed PRS using either Japanese or European GWAS data alone (Koyama et al., 2020). Similarly, it has been demonstrated that genetic data from African ancestry (both continental and diaspora groups) may enhance PRS performance for lipid traits in sub-Saharan Africans (Graham et al., 2021; Choudhury et al., 2022; Kamiza et al., 2022). Moreover, the consideration of Africa as a homogenous group in PRS evaluation might, at times, be misleading in cases as for lipid traits, as the same PRS model might have very different performance in different African geographic regions (Graham et al., 2021; Kamiza et al., 2022). T2DM PRSs have been widely developed in European populations (Vassy et al., 2012a; Walford et al., 2012) with evidence of high predictive utility beyond that of established risk factors, yet other populations experience higher rates of T2DM incidence. Trans-ancestry PRSs have recently been constructed for T2DM, integrating data from European, African, Hispanic and East Asian populations, with the top 2% of this PRS distribution identifying individuals with a 2.5–4.5-fold increased risk of developing T2D (Ge et al., 2022). A major limitation of the clinical utility of PRS in diverse populations is uncertainty in how best to accurately combine multi-ancestry GWAS data. Trans-ancestry PRSs do not incorporate population-specific allele frequency and LD patterns, and training PRS separately in each ancestry is complicated by discrepancies between self-reported ethnicity and genetic ancestry (Wilson et al., 2001). These limitations can be addressed by expanding data sources of non-European ancestries and conducting larger GWASs in these populations.

## Mendelian randomisation studies

In this review, we previously described ancestral differences in cardiometabolic risk factors and CVD incidence that have been explored in observational settings. However, observational data are prone to confounding and reverse causation, which limits the ability to make causal inferences about the role of risk factors in CVD occurrence and progression. Mendelian randomisation (MR) studies help overcome these limitations by using genetic variants as proxies for exposures (risk factors) to study their effects on outcomes (diseases). Given the relative paucity of GWAS data in non-European populations, relatively few MR studies have been conducted in non-European ancestries, thereby hindering our understanding of the causal role of risk factors in disease pathogenesis in different ancestries. Yet, findings from ancestry-specific MR studies can provide substantial insights into disease mechanisms. This is exemplified in a recent study by Fatumo et al. (2021), who investigated the causal effects of T2DM liability and lipid traits on ischaemic stroke risk in African ancestry populations. Their findings highlighted causal effects of T2DM and lipid traits on stroke risk for African ancestry individuals, the estimates of which were similar in European populations. Similarly, Soremekun et al. (2022) investigated the relationship between dyslipidaemia and T2DM in African ancestry individuals. Zheng et al. (2022) showed that the causal relationship between cardiometabolic risk factors and chronic kidney disease (CKD) may vary between Europeans and East Asian ancestry individuals. While eight cardiometabolic risk factors, including BMI, T2DM, nephrolithiasis and lipid biomarkers, showed causal effects on CKD in Europeans, only BMI, T2DM and nephrolithiasis showed evidence of causality in East Asians. It remains unclear, however, how much of this discrepancy can be explained by varying statistical power available for analyses across ancestry groups.

## Gene–environment interactions

Disease pathogenesis is a result of the interactions between information coded in the DNA and environmental factors (Zerba and Sing, 1993). Gene–environment interactions exist for almost every polygenic disease, including CVD (Ordovas and Shen, 2008; Andreasen and Andersen, 2009; Andreassi, 2009; Hirvonen, 2009). The study of gene–environment interactions can provide additional insight into disease pathogenesis and can help determine the public health impact of risk factors, thus informing public health policy (Zerba et al., 1996, 2000). Accounting for gene–environment interactions in GWASs can improve our understanding of genetic heterogeneity under different environmental exposures (Zhao et al., 2015). To identify adiposity loci whose effects are mediated by physical activities, Graff et al. (2017) undertook a meta-analysis of BMI and BMI-adjusted waist circumference and waist–hip ratio in Europeans and non-European individuals. They found an interaction with physical activity and *FTO* gene, and also discovered 11 novel loci for adiposity. As another example, Hindy et al. (2014) found that the increased risk of CVD mediated by rs4977574 is modified by vegetable and wine intake.

## Fairness, bias and future perspectives

As recently as 2019, it was estimated that 72% of GWAS participants were recruited in just three countries: the United States, the United Kingdom and Iceland (Peterson et al., 2019) Accordingly, there is an imperative to increase representation of non-European ancestries in large cross-ancestry GWAS' conducted in Europe and North America, and to conduct large-scale CVD GWAS' in developing countries, where the age-standardised death rate attributable to CVD is increasing rapidly (Roth et al., 2020). Academics in lower resource settings must be empowered if we are to seriously address such stark selection bias. Greater collaboration between those institutions in which large-scale genomic methods are most established and those institutions best placed to recruit underrepresented populations will be critical. The allocation of research funding should also give more explicit consideration to ancestry-related disparities in recruitment.

As outlined in this review, an inadequate understanding of genetic heterogeneity across ancestries may exacerbate existing inequalities in CVD outcomes. A lack of appreciation of differences in drug responsiveness may lead to individuals from certain ancestries being prescribed less effective medications. The absence of diverse ancestry information in PRSs can lead to poor prediction of disease in non-European ancestry populations. The preferential application of novel genetic methods such as MR in European ancestry populations could lead to the licencing of treatments for which the evidence base in other ancestries is extrapolative, uncertain and ultimately less efficacious or safe.

We illustrate some of these considerations with the following example. Hypertension is much more prevalent in African ancestry individuals (Spence and Rayner, 2018). According to the NICE guidelines, first-line antihypertensive agents differ for African ancestry individuals, with a preference for calcium channel blockers or diuretics over ACE inhibitors, given the relatively weaker response to the latter (Sinnott et al., 2020). The risk of stroke in hypertensive African ancestry individuals is three times greater than that of hypertensive Europeans (Spence and Rayner, 2018). An absence of any appreciation of genetic heterogeneity across ancestries could easily result in, and indeed may partly explain, the well-documented disparity in stroke outcomes between African ancestry and European ancestry individuals (Stansbury et al., 2005) Conversely, an understanding of such genetic heterogeneity can produce an appropriately higher index of suspicion of hypertension in African ancestry individuals, a tailored approach to treating and managing their hypertension and ultimately and an improvement in stroke-related disability in such individuals. Furthermore, genes such as LRP1, which are selectively associated with stroke in African ancestry individuals, could be both incorporated to improve stroke PRS and investigated as a novel target for the treatment of stroke, specifically in African ancestry individuals.

## Conclusions

In this review, we highlight and discuss the growing appreciation of genetic heterogeneity across ancestries in the development and progression of CVD. By elucidating such heterogeneity, we can better identify those molecular mechanisms that are common across different ancestries and those that are specific to certain ancestries. An understanding of such heterogeneity can facilitate the practice of precision medicine in three key ways. First, we can incorporate such heterogeneity to improve the clinical utility of PRSs in population risk stratification and primary prevention of CVD. Second, we can better understand how ancestry can produce differences in drug responsiveness, which can inform prescribing practises. Third, we can leverage tools such as MR to therapeutically target those mechanisms causally driving CVD both within and across ancestries.

**Open peer review.** To view the open peer review materials for this article, please visit http://doi.org/10.1017/pcm.2022.13.

**Data availability statement.** All data used are publicly available and cited in the article.

**Acknowledgement.** The authors would like to thank Ananyo Choudhury for providing helpful comments on an earlier draft of this review.

**Author contributions.** O.S., M.-J.D., S.R. and D.G. drafted the manuscript. All authors revised the manuscript for intellectual content. All authors approved the final version.

**Financial support.** O.S. is supported by the Africa Research Excellence Fund (AREF-325-SORE-F-C0904). S.F. is supported by the Wellcome Trust grant (220740/Z/20/Z) at the MRC/UVRI and LSHTM. D.G. is supported by the British Heart Foundation Centre of Research Excellence at Imperial College London (RE/18/4/34215).

**Competing interest.** D.G. is employed part-time by Novo Nordisk outside the submitted work. The remaining authors declare no relevant competing interest.

**Ethics standards.** This review article is based on published work, and no ethical approval was sought.

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
