## [Reviewer Report]

*Comments to Author*: This article gives and overview how CVD and potential mechanism for CVD differ by ancestry due to genetic heterogeneity. The authors also describe how this genetic heterogeneity is being utilized to inform the development of potential drug development and therapeutic interventions. The role of genetics in the drug development process are illustrated using exemplars of polygenetic risk scores and Mendelian Randomization studies for CAD and stroke and their modifiable risk factors, such as BMI and lipids in different ancestry groups.

1. On page 5 the authors make the point that “Whilst there are other CVD endpoints and modifiable risk factors, these few were selected based on their burden and contributions to CVD.” This seems reasonable but it was not clear whether there was a systematic search strategy utilized for the studies included in the review. It would be better if this was the case as it seems as if studies have been cherry-picked. I would like to see the search strategy reported with clear inclusion and exclusion criteria for a scoping review or something similar.

2. The authors mention development of therapeutic targets for ALDH2 for alcohol use disorder. This is interesting but wasn’t there a planned trial targeting AUD using HORIZANT in extended-release tablets that was unable to be conducted due to ethical issues. Would Antabuse potentially be limited by the same issues?

3. The paper by Sun et al. (Nature Medicine https://doi.org/10.1038/s41591-019-0366-x ) that showed concordance between observational, Mendelian Randomization and RCT data for the opposing relationship of LDL-c with ischaemic and haemorrhagic stroke in a Chinese population should be included as it provides good evidence for benefits and harm for LDL lowering for CHD and stroke.

4. I would have liked to see a section on “fairness and bias” included in the review with recommendations on how the representation of different ancestries in CVD precision medicine research can be improved as well as strategies to empower researchers from low resource settings. Also how the findings of research will be used is also important as this could exacerbate inequalities and bias against different ancestries (e.g. differences in drug responses).

5. Table 2 reports identical “ancestral differences” for CCB and Diuretics. This seems to be an error.

6. There are several typographical errors throughout the manuscript. For example on page 5 the authors wrote “…it also allude to an individual’s phenotype…“ should be “it also alludes to an individual’s phenotype…”. On page 6 “with established risk facts….” Should be “…established risk factors…”

---

## [Reviewer Report]

*Comments to Author*: The authors have responded satisfactorily to all my comments and I have no further comments to make.